# Robust Reinforcement Learning on State Observations with Learned Optimal Adversary

**Huan Zhang**[*,1]    **Hongge Chen**[*,2]    **Duane Boning**[2]    **Cho-Jui Hsieh**[1]

[1]Department of Computer Science, UCLA    [2]Department of EECS, MIT
huan@huan-zhang.com, chenhg@mit.edu, boning@mtl.mit.edu
chohsieh@cs.ucla.edu
[*]Huan Zhang and Hongge Chen contributed equally.

## Abstract

We study the robustness of reinforcement learning (RL) with adversarially perturbed state observations, which aligns with the setting of many adversarial attacks to deep reinforcement learning (DRL) and is also important for rolling out real-world RL agent under unpredictable sensing noise. With a fixed agent policy, we demonstrate that an *optimal adversary* to perturb state observations can be found, which is guaranteed to obtain the worst case agent reward. For DRL settings, this leads to a novel empirical adversarial attack to RL agents via a *learned* adversary that is much stronger than previous ones. To enhance the robustness of an agent, we propose a framework of alternating training with learned adversaries (ATLA), which trains an adversary online together with the agent using policy gradient following the optimal adversarial attack framework. Additionally, inspired by the analysis of state-adversarial Markov decision process (SA-MDP), we show that past states and actions (history) can be useful for learning a robust agent, and we empirically find a LSTM based policy can be more robust under adversaries. Empirical evaluations on a few continuous control environments show that ATLA achieves state-of-the-art performance under strong adversaries. Our code is available at https://github.com/huanzhang12/ATLA_robust_RL.

## 1 Introduction

Modern deep reinforcement learning agents (Mnih et al., 2015; Levine et al., 2015; Lillicrap et al., 2015; Silver et al., 2016; Fujimoto et al., 2018) typically use neuron networks as function approximators. Since the discovery of adversarial examples in image classification tasks (Szegedy et al., 2013), the vulnerabilities in DRL agents were first demonstrated in (Huang et al., 2017; Lin et al., 2017; Kos & Song, 2017) and further developed under more environments and different attack scenarios (Behzadan & Munir, 2017a; Pattanaik et al., 2018; Xiao et al., 2019). These attacks commonly add imperceptible noises into the *observations of states*, e.g., the observed environment slightly differs from true environment. This raises concerns for using RL in safety-crucial applications such as autonomous driving (Sallab et al., 2017; Voyage, 2019); additionally, the discrepancy between ground-truth states and agent observations also contributes to the "reality gap" - an agent working well in simulated environments may fail in real environments due to noises in observations (Jakobi et al., 1995; Muratore et al., 2019), as real-world sensing contains unavoidable noise (Brooks, 1992).

We classify the weakness of a DRL agent on the perturbations of state observations into two classes: the *vulnerability in function approximators*, which typically originates from the highly non-linear and blackbox nature of neural networks; and *intrinsic weakness of policy*: even perfect features for states are extracted, an agent can still make mistakes due to an intrinsic weakness in its policy.

For example, in the deep Q networks (DQNs) for Atari games, a large convolutional neural network (CNN) is used for extracting features from input frames. To act correctly, the network must extract crucial features: e.g., for the game of Pong, the position and velocity of the ball, which can observed by visualizing convolutional layers (Hausknecht & Stone, 2015; Guo et al., 2014). Many attacks to the DQN setting add imperceptible noises (Huang et al., 2017; Lin et al., 2017; Kos & Song, 2017; Behzadan & Munir, 2017a) that exploit the vulnerability of deep neural networks so that they extract wrong features, as we have seen in adversarial examples of image classification tasks. On the other

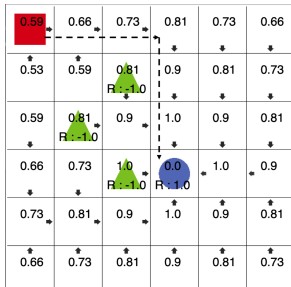
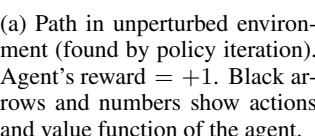
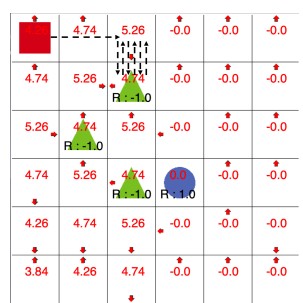
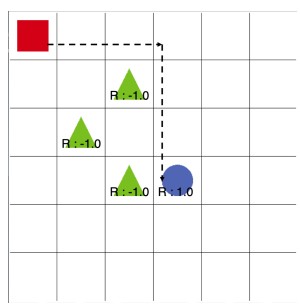

(a) Path in unperturbed environment (found by policy iteration). Agent's reward = +1. Black arrows and numbers show actions and value function of the agent.

(b) Path under the *optimal adversary*. Agent's reward = −∞. Red arrows and numbers show actions and value function of the *optimal* adversary (Section 3.1).

(c) A robust POMDP policy solved by SARSOP (Kurniawati et al., 2008) under the same adversary. This policy is *history dependent* (Section 3.2).

Figure 1: We show an agent in gridworld environment trained with no function approximators, and its optimal policy is intrinsically not robust to perturbations of state observations. The red square and blue circle are the starting point and target (reward +1) of the agent, respectively. The green triangles are traps, with reward -1 once encountered. The adversary is allowed to perturb the observation to adjacent states along four directions: up, down, left, and right. Adversary earns +1 at traps and -1 at the target. We set $\gamma = 0.9$ for both agent and adversary. This example shows that the vulnerability of a RL agent does not only come from the errors in function approximators such as DNNs.

hand, the fragile function approximation is not the only source of the weakness of a RL agent - in a finite-state Markov decision process (MDP), we can use tabular policy and value functions so there is no function approximation error. The agent can still be vulnerable to small perturbations on observations, e.g., perturbing the observation of a state to one of its four neighbors in a gridworld-like environment can prevent an agent from reaching its goal (Figure 1). To improve the robustness of RL, we need to take measures from both aspects — a more robust function approximator, and a policy aware of perturbations in observations.

Techniques developed in enhancing the robustness of neural network (NN) classifiers can be applied to address the vulnerability in function approximators. Especially, for environments like Atari games with images as input and discrete actions as outputs, the policy network $\pi_\theta$ behaves similarly to a classifier in test time. Thus, Fischer et al. (2019); Mirman et al. (2018a) utilized existing certified adversarial defense (Mirman et al., 2018b; Wong & Kolter, 2018; Gowal et al., 2018; Zhang et al., 2020a) approaches in supervised learning to enhance the robustness of DQN agents. Another successful approach (Zhang et al., 2020b) for both Atari and high-dimensional continuous control environment regularizes the smoothness of the learned policy such that $\max_{\hat{s} \in \mathcal{B}(s)} D(\pi_\theta(s), \pi_\theta(\hat{s}))$ is small for some divergence $D$ and $\mathcal{B}(s)$ is a neighborhood around $s$. This maximization can be solved using a gradient based method or convex relaxations of NNs (Salman et al., 2019; Zhang et al., 2018; Xu et al., 2020), and then minimized by optimizing $\theta$. Such an adversarial minimax regularization is in the same spirit as the ones used in some adversarial training approaches for (semi-)supervised learning, e.g., TRADES (Zhang et al., 2019) and VAT (Miyato et al., 2015). However, regularizing the function approximators does not explicitly improve the intrinsic policy robustness.

In this paper, we propose an orthogonal approach, alternating training with learned adversaries (ATLA), to enhance the robustness of DRL agents. We focus on dealing with the intrinsic weakness of the policy by learning an adversary online with the agent during training time, rather than directly regularizing function approximators. Our main contributions can be summarized as:

• We follow the framework of state-adversarial Markov decision process (SA-MDP) and show how to learn an *optimal* adversary for perturbing observations. We demonstrate practical attacks under this formulation and obtain learned adversaries that are significantly stronger than previous ones.

• We propose the alternating training with learned adversaries (ATLA) framework to improve the robustness of DRL agents. The difference between our approach and previous adversarial training approaches is that we use a stronger adversary, which is learned online together with the agent.

• Our analysis on SA-MDP also shows that *history* can be important for learning a robust agent. We thus propose to use a LSTM based policy in the ATLA framework and find that it is more robust than policies parameterized as regular feedforward NNs.

• We evaluate our approach empirically on four continuous control environments. We outperform explicit regularization based methods in a few environments, and our approach can also be directly combined with explicit regularizations on function approximators to achieve state-of-the-art results.

## 2 RELATED WORK

State-adversarial Markov decision process (SA-MDP) (Zhang et al., 2020b) characterizes the decision making problem under adversarial attacks on *state observations*. Most importantly, the true state in the environment is not perturbed by the adversary under this setting; for example, perturbing pixels in an Atari environment (Huang et al., 2017; Kos & Song, 2017; Lin et al., 2017; Behzadan & Munir, 2017a; Inkawhich et al., 2019) does not change the true location of an object in the game simulator. SA-MDP can characterize agent performance under natural or adversarial noise from sensor measurements. For example, GPS sensor readings on a car are naturally noisy, but the ground truth location of the car is not affected by the noise. Importantly, this setting is different from robust Markov decision process (RMDP) (Nilim & El Ghaoui, 2004; Iyengar, 2005), where the worst case transition probabilities of the environment are considered. "Robust reinforcement learning" in some works (Mankowitz et al., 2018; 2019) refer to this different definition of robustness in RMDP, and should not be confused with our setting of robustness against perturbations on state observations.

Several works proposed methods to learn an adversary online together with an agent. RARL (Pinto et al., 2017) proposed to train an agent and an adversary under the two-player Markov game (Littman, 1994) setting. The adversary *can* change the environment states through actions *directly applied* to environment. The goal of RARL is to improve the robustness against environment parameter changes, such as mass, length or friction. Gleave et al. (2019) discussed the learning of an adversary using reinforcement learning to attack a victim agent, by taking adversarial actions that changes the environment and consequentially change the observation of the victim agent. Both Pinto et al. (2017); Gleave et al. (2019) conduct their attack under on the two-player Markov game framework, rather than considering perturbations on state observations. Besides, Li et al. (2019) consider a similar Markov game setting in multi-agent RL environments. The difference between these works and ours can be clearly seen in the setting where the adversary is fixed - under the framework of (Pinto et al., 2017; Gleave et al., 2019), the learning of agent is still a MDP, but in our setting, it becomes a harder POMDP problem (Section 3.2).

Training DRL agents with perturbed state observations from adversaries have been investigated in a few works, sometimes referred to as adversarial training. Kos & Song (2017); Behzadan & Munir (2017b) used gradient based adversarial attacks to DQN agents and put adversarial frames into replay buffer. This approach is not very successful because for Atari environments the main source of weakness is likely to come from the function approximator, so an adversarial regularization framework such as (Zhang et al., 2020b; Qu et al., 2020) which directly controls the smoothness of the $Q$ function is more effective. For lower dimensional continuous control tasks such as the MuJoCo environments, Mandlekar et al. (2017); Pattanaik et al. (2018) conducted FGSM and multi-step gradient based attacks during training time; however, their main focus was on the robustness against environment parameter changes and only limited evaluation on the adversarial attack setting was conducted with relatively weak adversaries. Zhang et al. (2020b) systematically tested this approach under newly proposed strong attacks, and found that it cannot reliably improve robustness. These early adversarial training approaches typically use gradients from a critic function. They are usually relatively weak, and not sufficient to lead to a robust policy under stronger attacks.

The robustness of RL has also been investigated from other perspectives. For example, Tessler et al. (2019) study MDPs under action perturbations; Tan et al. (2020) use adversarial training on action space to enhance agent robustness under action perturbations. Besides, policy teaching (Zhang & Parkes, 2008; Zhang et al., 2009; Ma et al., 2019) and policy poisoning (Rakhsha et al., 2020; Huang & Zhu, 2019) manipulate the reward or cost signal during agent training time to induce a desired agent policy. Essentially, policy teaching is a training time "attack" with *perturbed rewards* from the environments (which can be analogous to data poisoning attacks in supervised learning settings), while our goal is to obtain a robust agent against test time adversarial attacks. All these settings differ from the setting of perturbing state observations discussed in our paper.

## 3 METHODOLOGY

In this section, we first discuss the case where the agent policy is fixed, and then the case where the adversary is fixed in SA-MDPs. This allows us to propose an alternating training framework to improve robustness of RL agents under perturbations on state observations.

**Notations and Background** We use $\mathcal{S}$ and $\mathcal{A}$ to represent the state space and the action space, respectively; $\mathcal{P}(\mathcal{S})$ defines the set of all possible probability measures on $\mathcal{S}$. We define a Markov decision process (MDP) as $(\mathcal{S}, \mathcal{A}, R, p, \gamma)$, where $R : \mathcal{S} \times \mathcal{A} \times \mathcal{S} \to \mathbb{R}$ and $p : \mathcal{S} \times \mathcal{A} \to \mathcal{P}(\mathcal{S})$ are two mappings represent the reward and transition probability. The transition probability at time step $t$ can be written as $p(s'|s, a) = \Pr(s_{t+1} = s'|s_t = s, a_t = a)$. Reward function is defined as the expected reward $R(s, a, s') := \mathbb{E}[r_t|s_t = s, a_t = a, s_{t+1} = s']$. $\gamma \in [0, 1]$ is the discounting factor. We denote a stationary policy as $\pi : \mathcal{S} \to \mathcal{P}(\mathcal{A})$ which is independent of history. We denote history $h_t$ at time $t$ as $\{s_0, a_0, \cdots, s_{t-1}, a_{t-1}, s_t\}$ and $\mathcal{H}$ as the set of all histories. A history-dependent policy is defined as $\pi : \mathcal{H} \to \mathcal{P}(\mathcal{A})$. A partially observable Markov decision process (Astrom, 1965) (POMDP) can be defined as a 7-tuple $(\mathcal{S}, \mathcal{A}, \mathcal{O}, \Omega, R, p, \gamma)$ where $\mathcal{O}$ is a set of observations and $\Omega$ is a set of conditional observation probabilities $p(o|s)$. Unlike MDPs, POMDPs typically require history-dependent optimal policies.

To study the decision problem under adversaries on state observations, we use state-adversarial Markov decision process (SA-MDP) framework (Zhang et al., 2020b). In SA-MDP, an adversary $\nu : \mathcal{S} \to \mathcal{P}(\mathcal{S})$ is introduced to perturb the input state of an agent; however, the true environment state $s$ is unchanged (Figure 2). Formally, an SA-MDP is a 6-tuple $(\mathcal{S}, \mathcal{A}, \mathcal{B}, R, p, \gamma)$ where $\mathcal{B}$ is a mapping from a state $s \in \mathcal{S}$ to a set of states $\mathcal{B}(s) \in \mathcal{S}$. The agent sees the perturbed state $\hat{s} \sim \nu(\cdot|s)$ and takes the action $\pi(a|\hat{s})$ accordingly. $\mathcal{B}$ limits the power of adver-

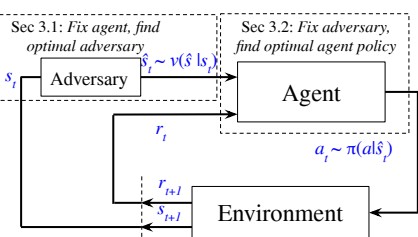

Figure 2: SA-MDP introduces an adversary on state observations in a MDP.

sary: supp $(\nu(\cdot|s)) \in \mathcal{B}(s)$. The goal of SA-MDP is to solve an optimal policy $\pi^*$ under its optimal adversary $\nu^*(\pi^*)$; an optimal adversary is defined as $\nu^*(\pi)$ such that $\pi$ achieves the lowest possible expected discounted return (or value) on all states. Zhang et al. (2020b) did not give an explicit algorithm to solve SA-MDP and found that a stationary optimal policy need not exist.

## 3.1 FINDING THE OPTIMAL ADVERSARY UNDER A FIXED POLICY

In this section, we discuss how to find an *optimal* adversary $\nu$ for a given policy $\pi$. An optimal adversary leads to the *worst case performance* under bounded perturbation set $\mathcal{B}$, and is an absolute lower bound of the expected cumulative reward an agent can receive. It is similar to the concept of "minimal adversarial example" in supervised learning tasks. We first show how to solve the optimal adversary in MDP setting and then apply it to the DRL settings.

A technical lemma (Lemma 1) from Zhang et al. (2020b) shows that, from the adversary's point of view, a fixed and stationary agent policy $\pi$ and the environment dynamics can be essentially merged into an MDP with redefined dynamics and reward functions:

**Lemma 1 (Zhang et al. (2020b))** *Given an SA-MDP $M = (\mathcal{S}, \mathcal{A}, R, \mathcal{B}, p, \gamma)$ and a fixed and stationary policy $\pi(\cdot|\cdot)$, there exists an MDP $\hat{M} = (\mathcal{S}, \hat{\mathcal{A}}, \hat{R}, \hat{p}, \gamma)$ such that the optimal policy of $\hat{M}$ is the optimal adversary $\nu$ for SA-MDP given the fixed $\pi$, where $\hat{\mathcal{A}} = \mathcal{S}$, and*

$$\hat{R}(s, \hat{a}, s') := \mathbb{E}[\hat{r}|s, \hat{a}, s'] = \begin{cases} -\frac{\sum_{a \in \mathcal{A}} \pi(a|\hat{a})p(s'|s,a)R(s,a,s')}{\sum_{a \in \mathcal{A}} \pi(a|\hat{a})p(s'|s,a)} & \text{for } s, s' \in \mathcal{S} \text{ and } \hat{a} \in \mathcal{B}(s) \subset \hat{\mathcal{A}}, \\ C & \text{for } s, s' \in \mathcal{S} \text{ and } \hat{a} \notin \mathcal{B}(s). \end{cases}$$

*where $C$ is a large negative constant, and*

$$\hat{p}(s'|s, \hat{a}) = \sum_{a \in \mathcal{A}} \pi(a|\hat{a})p(s'|s, a) \quad \text{for } s, s' \in \mathcal{S} \text{ and } \hat{a} \in \hat{\mathcal{A}}.$$

The intuition behind Lemma 1 is that the adversary's goal is to reduce the reward earned by the agent. Thus, when a reward $r_t$ is received by the agent at time step $t$, the adversary receives a negative reward of $\hat{r}_t = -r_t$. To prevent the agent from taking actions outside of set $\mathcal{B}(s)$, a large negative reward $C$ is assigned to these actions such that the optimal adversary cannot take them. For actions within $\mathcal{B}(s)$, we calculate $\hat{R}(s, \hat{a}, s')$ by its definition, $\hat{R}(s, \hat{a}, s') := \mathbb{E}[\hat{r}|s, \hat{a}, s']$ which yields the term in Lemma 1. The proof can be found in Appendix B of Zhang et al. (2020b).

After constructing the MDP $\hat{M}$, it is possible to solve an optimal agent $\nu$ of $\hat{M}$, which will be the optimal adversary on SA-MDP $M$ given policy $\pi$. For MDPs, under mild regularity assumptions an

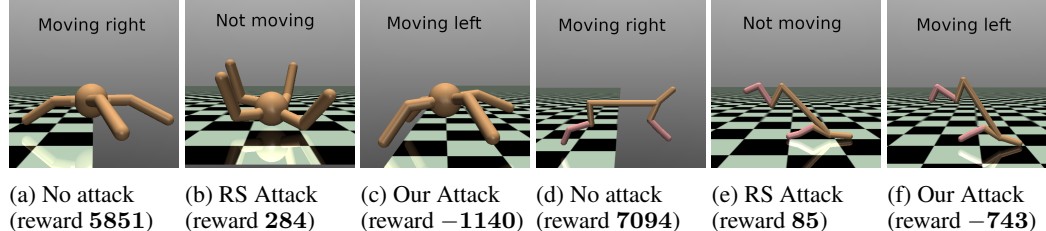

(a) No attack
(reward **5851**)

(b) RS Attack
(reward **284**)

(c) Our Attack
(reward −**1140**)

(d) No attack
(reward **7094**)

(e) RS Attack
(reward **85**)

(f) Our Attack
(reward −**743**)

Figure 3: Our "Optimal" Attack and Robust Sarsa attack (a previous *strong* attack proposed in Zhang et al. (2020b)) on Ant and HalfCheetah environments. Previous strong attacks make the agent fail and receive a small positive reward (less than 1/10 of the reward without attack). Our attack is strong enough to trick the agent into moving to the opposite direction, receiving a large *negative reward*.

optimal policy always exists (Puterman, 2014). In our case, the optimal policy on $\hat{M}$ corresponds to an *optimal adversary* in SA-MDP, which is the worst case perturbation for policy $\pi$. As an illustration, in Figure 1, we show a GridWorld environment. The red square is the starting point. The blue circle and green triangles are the target and traps, respectively. When the agent hits the target, it earns reward +1 and the game stops and it earns reward -1 when it encounters a trap. We set $\gamma = 0.9$ for both agent and adversary. The adversary is allowed to perturb the observation to adjacent cells along four directions: up, down, left, and right. When there is no adversary, after running policy iteration, the agent can easily reach target and earn a reward of +1, as in Figure 1a. However, if we train the adversary based on Lemma 1 and apply it to the agent, we are able to make the agent repeatedly encounter a trap. This leads to $-\infty$ reward for the agent and $+\infty$ reward for the adversary as shown in Figure 1b.

We now extend this Lemma 1 to the DRL setting. Since the learning of adversary is equivalent to solving an MDP, we parameterize the adversary as a neural network function and use any popular DRL algorithm to learn an "optimal" adversary. Here we quote the word "optimal" as we use function approximator to learn the agent so it's no longer optimal, but we emphasize that it follows the SA-MDP framework of solving an optimal adversary. No existing adversarial attacks follow such a theoretically guided framework. We show our algorithm in Algorithm 1. Instead of learning to produce $\hat{s} \in \mathcal{B}(s)$ directly, since $\mathcal{B}(s)$ is usually a small set nearby $s$ (e.g., $\mathcal{B} = \{s' | \|s - s'\|_p \leq \epsilon\}$, our adversary learns a perturbation vector $\Delta$, and we project $s + \Delta$ to $\mathcal{B}(s)$.

The first advantage of attacking a policy in this way is that it is *strong* - as we allow to optimize the adversary in an online loop of interactions with the agent policy and environment, and keep improving the adversary with a goal of receiving as less reward as possible. It is strong because it follows the theoretical framework of finding an optimal adversary, rather than using any heuristic to generate a perturbation. Empirically, in the cases demonstrated in Figure 3, previous strong attacks (e.g., Robust Sarsa attack) can successfully fail an agent and make it stop moving and receive a small positive reward; our learned attack can trick the agent into *moving toward the opposite direction of the goal* and receive a large negative reward. We also find that this attack can further reduce the reward of robustly trained agents, like SA-PPO (Zhang et al., 2020b).

The second advantage of this attack is that it requires no gradient access to the policy itself; in fact, it treats the agent as part of the environment and only needs to run it in a blackbox. Previous attacks (e.g., Lin et al. (2017); Pattanaik et al. (2018); Xiao et al. (2019)) are mostly gradient based approach and need to access the values or gradients to a policy or value function. Even without access to gradients, the overall learning process is still just a MDP and we can apply any popular modern DRL methods to learn the adversary.

## 3.2 Finding the optimal policy under a fixed adversary

We now investigate SA-MDP when we fix the adversary $\nu$ and find an optimal policy. In Lemma 2, we show that this case SA-MDP becomes a POMDP:

**Lemma 2 (Optimal policy under fixed adversary)** *Given an SA-MDP $M = (\mathcal{S}, \mathcal{A}, \mathcal{B}, R, p, \gamma)$ and a fixed and stationary adversary $\nu(\hat{s}|s)$, there exists a POMDP $\bar{M} = (\mathcal{S}, \mathcal{A}, \Omega, O, R, p, \gamma)$*

---

**Algorithm 1** Learning an "optimal" adversary for perturbations on state observations

---

**Input:** Policy $\pi(\cdot|s)$ under attack, number of iterations $N_{iter}$, batch size $B$, perturbation set $\mathcal{B}(s)$
1: initialize adversary $\nu_\phi(\cdot|s)$ parameterized by a neural network with parameters $\phi$,
2: **for** $i = 1$ to $N_{iter}$ **do**
3:     $\mathcal{D} \leftarrow Adv\_Traj(\nu_\phi, \pi, B)$     # collection of samples (for simplicity we ignore episodes here)
4:     $\phi \leftarrow \text{PolicyOptimizer}(\mathcal{D}, \phi)$
5: **end for**

**Function** $Adv\_Traj(\nu_\phi, \pi, B)$ :
6: $s \leftarrow s_0$                                                       # Initial state
7: $\mathcal{D} \leftarrow \emptyset$
8: **for** $b = 1$ to $B$ **do**
9:     $\Delta \leftarrow$ sample from $\nu_\phi(\cdot|s)$
10:     $\hat{s} \leftarrow \text{Proj}_{\mathcal{B}}(s)(s + \Delta)$                          # projection will be a clipping for $\ell_\infty$ norm set $\mathcal{B}(s)$
11:     $a \leftarrow$ sample from $\pi(\cdot|\hat{s})$
12:     obtain current step reward $r_t$, next state $s'$ from environment given action $a$
13:     $\mathcal{D} \leftarrow \mathcal{D} \cup (s, \Delta, -r_t, s')$          # state, action, reward and next state for the adversary
14:     $s \leftarrow s'$
15: **end for**
16: **return** $\mathcal{D}$

---

*such that the optimal policy of $\bar{M}$ is the optimal policy $\pi$ for SA-MDP given the fixed $\nu$, where*

$$\Omega = \bigcup_{s \in \mathcal{S}} \{s' | s' \in supp(\nu(\cdot|s))\}, \qquad O(o|s) = \nu(\hat{s}|s) \qquad (1)$$

where $\Omega$ is the set of observations, and $O$ defines the conditional observational probabilities (in our case it is conditioned only on $s$ and does not depend on actions). To prove Lemma 2, we construct the a POMDP with the observations defined on the support of all $\nu(\cdot|s), s \in \mathcal{S}$ and the observation process is exactly the process of generating an adversarially perturbed state $\hat{s}$. This POMDP is functionally identical to the original SA-MDP when $\nu$ is fixed. This lemma unveils the connection between POMDP and SA-MDP: SA-MDP can be seen as a version of "robust" POMDP where the policy needs to be robust under a set of observational processes (adversaries). SA-MDP is different from robust POMDP (RPOMDP) (Osogami, 2015; Rasouli & Saghafian, 2018), which optimizes for the worst case environment transitions.

As a proof of concept, we use a modern POMDP solver, SARSOP (Kurniawati et al., 2008) to solve the GridWorld environment in Figure 1 to find a policy that can defeat the adversary. The POMDP solver produces a finite state controller (FSC) with 8 states (FSC is an efficient representation of history dependent policies). This FSC policy can almost eliminate the impact of the adversary and receive close to perfect reward, as shown in Figure 1c.

Unfortunately, unlike MDPs, it is challenging to solve an optimal policy for POMDPs; state-of-the-art solvers (Bai et al., 2014; Sunberg & Kochenderfer, 2017) can only work on relatively simple environments which are much smaller than those used in modern DRL. Thus, we do not aim to solve the optimal policy. We follow (Wierstra et al., 2007) to use recurrent policy gradient theorem on POMDPs and use LSTM as function approximators for the value and policy networks. We denote $h_t = \{\hat{s}_0, a_0, \hat{s}_1, a_1 \cdots, \hat{s}_t\}$ containing all history of states (perturbed states $\hat{s}$ in our setting) and actions. The policy $\pi$ parameterized by $\theta$ takes an action $a_t$ given all observed history $h_t$, and $h_t$ is typically encoded by a recurrent neural network (e.g., LSTM). The recurrent policy gradient theorem (Wierstra et al., 2007) shows that

$$\nabla_\theta J \approx \frac{1}{N} \sum_{n=1}^{N} \sum_{t=0}^{T} \nabla_\theta \log \pi_\theta(a_t^n | h_t^n) r_t^n \qquad (2)$$

where $N$ is the number of sampled episodes, $T$ is episode length (for notation similarity, we assume each episode has the same length), and $h_t^n$ is the history of states for episode $n$ up to time $t$, and $r_t^n$ is the reward received for episode $n$ at time $t$. We can then extend Eq. 2 to modern DRL algorithms such as proximal policy optimization (PPO), similarly as done in (Azizzadenesheli et al., 2018), by

using the following loss function:

$$J(\theta) \approx \frac{1}{N} \sum_{n=1}^{N} \sum_{t=0}^{T} \left[ \min \left( \frac{\pi_\theta(a_t^n | h_t^n)}{\pi_{\theta_{\text{old}}}(a_t^n | h_t^n)} A_{h_t^n}, \text{clip}(\frac{\pi_\theta(a_t^n | h_t^n)}{\pi_{\theta_{\text{old}}}(a_t^n | h_t^n)}, 1 - \epsilon, 1 + \epsilon) A_{h_t^n} \right) \right] \quad (3)$$

where $A_{h_t^n}$ is a baseline advantage function for episode $n$ time step $t$, which is based on a LSTM value function. $\epsilon$ is the clipping threshold in PPO. The loss can be optimized via a gradient based optimizer and $\theta_{\text{old}}$ is the old policy parameter before optimization iterations start. Although a LSTM or recurrent policy network has been used in the DRL setting in a few other works (Hausknecht & Stone, 2015; Azizzadenesheli et al., 2018), our focus is to improve agent robustness rather than learning a policy purely for POMDPs. In our empirical evaluation, we will compare feedforward and LSTM policies under our ATLA framework.

### 3.3 ALTERNATING TRAINING WITH LEARNED ADVERSARIES (ATLA)

As we have discussed in Section 3.1, we can solve an optimal adversary given any fixed policy. In our ATLA framework, we train such an adversary online with the agent: we first keep the agent and optimize the adversary; the adversary is also parameterized as a neural network. Then we keep the adversary and optimize the agent. Both adversary and agent can be updated using a policy gradient algorithm such as PPO. We show our full algorithm in Algorithm 2.

---

**Algorithm 2** Alternating Training with Learned Adversaries (ATLA)

---

**Input:** Environment $\mathcal{E}$, number of iterations $N_{iter}$, and batch size $B$.
1: Initialize the agent's actor network $\pi(a|\hat{s})$ with parameters $\theta$.
2: Initialize the adversary's actor network $\nu(\hat{s}|s)$ with parameters $\phi$.
3: **for** $i = 1$ to $N_{iter}$ **do**
4:     **for** $j = 1$ to $N_\pi$ **do**
5:         Run $\pi_\theta$ with fixed $\nu_\phi$ to collect a set of trajectories $\mathcal{D}_\pi := \{(\hat{s}_t^{k,j}, a_t^{k,j}, r_t^{k,j}, \hat{s}_{t+1}^{k,j})\}\big|_{k=1}^{B}$.
6:         $\theta \leftarrow \text{PolicyOptimizer}(\mathcal{D}_\pi, \theta)$
7:     **end for**
8:     **for** $j = 1$ to $N_\nu$ **do**
9:         $\mathcal{D}_\nu \leftarrow Adv\_Traj(\nu_\phi, \pi_\theta, B)$          # *Adv_Traj* defined in Algorithm 1
10:         $\phi \leftarrow \text{PolicyOptimizer}(\mathcal{D}_\nu, \phi)$
11:     **end for**
12: **end for**

---

Our algorithm is designed to use a strong and learned adversary that tries to find intrinsic weakness of the policy, and to obtain a good reward the policy must learn to defeat such an adversary. In other words, it attempts to solve the SA-MDP problem directly rather than relying on explicit regularization on the function approximator like the approach in (Zhang et al., 2020b). In our empirical evaluation, we show that such regularization can be unhelpful in some environments and harmful for performance when evaluating the agent without attacks.

The difference between our approach and previous adversarial training approaches such as (Pattanaik et al., 2018) is that we use a stronger adversary, learned online with the agent. Our empirical evaluation finds that using such a learned "optimal" adversary in training time allows the agent to learn a robust policy generalized to different types of strong adversarial attacks during test time. Additionally, it is important to distinguish between the original state $s$ and the perturbed state $\hat{s}$. We find that using $s$ instead of $\hat{s}$ to train the advantage function and policy of the agent leads to worse performance, as it does not follow the theoretical framework of SA-MDP.

## 4 EXPERIMENTS

**"Optimal" attack on DRL agents**[1] In section 3.1 we show that it is possible to cast the optimal adversary finding problem as an MDP problem. In practice, the environment dynamics are unknown but model-free RL methods can be used to approximately find this optimal adversary. In this section, we use PPO to train an adversary on four OpenAI Gym MuJoCo continuous control environments.

---

[1]Code for the optimal attack and ATLA available at `https://github.com/huanzhang12/ATLA_robust_RL`

Table 1: Average episode rewards ± standard deviation over 50 episodes on PPO and SA-PPO agents. We report natural rewards (no attacks) and rewards under six adversarial attacks, including a simple random noise attack, the critic based attack in Pattanaik et al. (2018),MAD and RS attacks in Zhang et al. (2020b), Snooping attack proposed in Inkawhich et al. (2019), and the optimal attack proposed in this paper. In each row we bold the best (lowest) attack reward over all five attacks. "Optimal" attack is better than other attacks in all environments, sometimes by a large margin.

| Env. | $\ell_\infty$ norm perturbation budget $\epsilon$ | Method | Natural Reward | Attack Reward | | | | | |
| | | | | Critic | Random | MAD | Snooping | RS | "Optimal" |
| Hopper | 0.075 | PPO | 3167±521 | 1464 ±523 | 2101±793 | 1410± 655 | 2234±1103 | 794±238 | **636± 9** |
| | | SA-PPO | 3705± 2 | 3789± 15 | 2710± 801 | 2652± 835 | 2509±838 | 1130 ±42 | **1076± 791** |
| Walker2d | 0.05 | PPO | 4472 ± 635 | 3424 ± 1295 | 3007 ± 1200 | 2869 ± 1271 | 2786±962 | 1336 ± 654 | **1086±516** |
| | | SA-PPO | 4487± 61 | 4875± 30 | 4867± 39 | 3668± 1789 | 3928±1661 | 3808± 138 | **2908± 1136** |
| Ant | 0.15 | PPO | 5687 ± 758 | 4934± 1022 | 5261± 1005 | 1759± 828 | 3668±547 | 268 ±227 | **-872 ± 436** |
| | | SA-PPO | 4292± 384 | 4805 ± 128 | 4986 ±452 | 4662 ±522 | 4079±768 | 3412 ±1755 | **2511 ± 1117** |
| HalfCheetah | 0.15 | PPO | 7117± 98 | 5761±119 | 5486± 1378 | 1836± 866 | 1637±843 | 489± 758 | **-660± 219** |
| | | SA-PPO | 3632± 20 | 3589± 21 | 3619± 18 | 3624± 23 | 3616±21 | 3283± 20 | **3028 ±23** |

Table 1 presents results on attacking vanilla PPO and robustly trained SA-PPO (Zhang et al., 2020b) agents. As a comparison, we also report the attack reward of five other baseline attacks: critic attack is based on (Pattanaik et al., 2018); random attack adds uniform random noise to state observations; MAD (maximal action difference) attack (Zhang et al., 2020b) maximizes the differences in action under perturbed states; RS (robust sarsa) attack is based on training robust action-value functions and is the strongest attack proposed in (Zhang et al., 2020b). Additionally, we include the black-box Snooping attack (Inkawhich et al., 2019). For all attacks we consider $\mathcal{B}(s)$ as a $\ell_\infty$ norm ball around $s$ with radius $\epsilon$, set similarly as in (Zhang et al., 2020b). During testing, we run the agents without attacks as well as under attacks for 50 episodes and report the mean and standard deviation of episode rewards. In Table 1 our "optimal" attack achieves noticeably lower rewards than all the other five attacks. We illustrate a few examples of attacks in Figure 3. For RS and "optimal" attacks, we report the best (lowest) attack reward obtained from different hyper-parameters.

**Evaluation of ATLA**  In this experiment, we study the effectiveness of our proposed ATLA method. Specifically, we use PPO as our policy optimizer. For policy networks, we have two different structures: the original fully connected (MLP) structure, and an LSTM structure which takes historical observations. The LSTMs are trained using backpropagation through time for up to 100 steps. In Table 2 we include the following methods for comparisons:

• PPO (vanilla) and PPO (LSTM): PPO with a feedforward NN or LSTM as the policy network.

• SA-PPO (Zhang et al., 2020b): the state-of-the-art approach for improving the robustness of DRL in continuous control environments, using a smooth policy regularization on feedforward NNs solved by convex relaxations.

• Adversarial training using critic attack (Pattanaik et al., 2018): a previous work using critic based attack to generate adversarial observations in training time, and train a feedforward NN based agent with this relatively weak adversary.

• ATLA-PPO (MLP) and ATLA-PPO (LSTM): Our proposed method trained with a feedforward NN (MLP) or LSTM as the policy network. The agent and adversary are trained using PPO with independent value and policy networks. For simplicity, we set $N_\pi = N_\nu = 1$ in all settings.

• ATLA-PPO (LSTM) +SA reg: Based on ATLA-PPO (LSTM), but with an extra adversarial smoothness constraint similar to those in SA-PPO. We use a 2-step stochastic gradient Langevin dynamics (SGLD) to solve the minimax loss, as convex relaxations of LSTMs are expensive.

For each agent, we report its "natural reward" (episode reward without attacks) and best attack reward in Table 2. To comprehensively evaluate the robustness of agents, the best attack reward is the *lowest* episode reward achieved by all six types attacks in Table 1, including our new "optimal" attack (these attacks include hundreds of independent adversaries for attacking a single agent, see Appendix A.1 for more details). For reproducibility, for each setup we train 21 agents, attack all of them and report the one with *median* robustness. We include detailed hyperparameters in A.5.

In Table 2 we can see that vanilla PPO with MLP or LSTM are not robust. For feedforward (MLP) agent policies, critic based adversarial training (Pattanaik et al., 2018) is not very effective under our suite of strong adversaries and is sometimes only slightly better than vanilla PPO. ATLA-PPO (MLP) outperforms SA-PPO on Hopper and Walker2d and is also competitive on HalfCheetah; for high dimensional environments like Ant, the robust function approximator regularization in SA-PPO is more effective. For LSTM agent policies, compared to vanilla PPO (LSTM) agents,

Table 2: Average episode rewards ± standard deviation over 50 episodes on ATLA agents and baselines. We report natural rewards (no attacks) and the best (lowest) attack rewards among six types of adversarial attacks, including a simple random noise attack, the critic based attack in (Pattanaik et al., 2018), MAD and RS attacks in Zhang et al. (2020b), Snooping attack proposed in Inkawhich et al. (2019), and the optimal attack proposed in this paper. For each environment, we bold the most robust agent. Since both RS attack and our "optimal" attack are parameterized attacks, the "best attack" column represents the worst case agent performance *under hundreds of adversaries*. See Appendix A.1 for more details.

| Env. | State Dimension | $\ell_\infty$ norm perturbation budget $\epsilon$ | Method | Natural Reward | Best Attack |
|---|---|---|---|---|---|
| Hopper | 11 | 0.075 | PPO (vanilla) | 3167±542 | 636± 9 |
| | | | SA-PPO (Zhang et al., 2020b) | 3705± 2 | 1076± 791 |
| | | | Pattanaik et al. (2018) | 2755±582 | 291± 7 |
| | | | ATLA-PPO (MLP) | 2559 ± 958 | 976± 40 |
| | | | PPO (LSTM) | 3060± 639.3 | 784± 48 |
| | | | ATLA-PPO (LSTM) | 3487± 452 | 1224± 191 |
| | | | **ATLA-PPO (LSTM) +SA Reg** | 3291± 600 | **1772± 802** |
| Walker2d | 17 | 0.05 | PPO (vanilla) | 4472 ± 635 | 1086±516 |
| | | | SA-PPO (Zhang et al., 2020b) | 4487± 61 | 2908± 1136 |
| | | | Pattanaik et al. (2018) | 4058± 1410 | 733± 1012 |
| | | | ATLA-PPO (MLP) | 3138 ± 1061 | 2213± 915 |
| | | | PPO (LSTM) | 2785± 1121 | 1259± 937 |
| | | | ATLA-PPO (LSTM) | 3920± 129 | 3219± 1132 |
| | | | **ATLA-PPO (LSTM) +SA Reg** | 3842± 475 | **3239± 894** |
| Ant | 111 | 0.15 | PPO (vanilla) | 5687 ± 758 | -872 ± 436 |
| | | | SA-PPO (Zhang et al., 2020b) | 4292± 384 | 2511 ± 1117 |
| | | | Pattanaik et al. (2018) | 3469± 1139 | -672± 100 |
| | | | ATLA-PPO (MLP) | 4894± 123 | 33±327 |
| | | | PPO (LSTM) | 5696 ± 165 | -513 ± 104 |
| | | | ATLA-PPO (LSTM) | 5612± 130 | 716± 256 |
| | | | **ATLA-PPO (LSTM) +SA Reg** | 5359±153 | **3765± 101** |
| HalfCheetah | 17 | 0.15 | PPO (vanilla) | 7117± 98 | -660± 218 |
| | | | SA-PPO (Zhang et al., 2020b) | 3632± 20 | 3028 ±23 |
| | | | Pattanaik et al. (2018) | 5241± 1162 | 447± 192 |
| | | | ATLA-PPO (MLP) | 5417± 49 | 2170± 2097 |
| | | | PPO (LSTM) | 5609± 98 | -886± 30 |
| | | | ATLA-PPO (LSTM) | 5766 ± 109 | 2485± 1488 |
| | | | **ATLA-PPO (LSTM) +SA Reg** | 6157± 852 | **4806± 603** |

ATLA-PPO (LSTM) can significantly improve agent robustness; a LSTM agent trained without a robust training procedure like ATLA cannot improve robustness. We find that LSTM agents tend to be more robust than their MLP counterparts, validating our findings in Section 3.2. ATLA-PPO (LSTM) is better than SA-PPO on Hopper and Walker2d. In all settings, especially for high dimensional environments like Ant, our ATLA approach that also includes State-Adversarial regularization (ATLA-PPO +SA Reg) outperforms all other baselines, as this combination improves both the intrinsic robustness of policy and the robustness of function approximator.

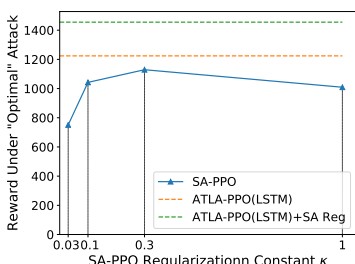

Figure 4: The performance under the strongest attack for SA-PPO Hopper with different regularization $\kappa$. Even we increase regularization, it cannot outperform our ATLA agents.

**A robust function approximator can be insufficient** For some environments, SA-PPO method has its limitations - even using an increasingly larger regularization parameter $\kappa$ (which controls how robust the function approximator needs to be), we still cannot reach the same performance as our ATLA agent (Figure 4). Additionally, when a large regularization is used, agent performance becomes much worse. In Figure 4, under the largest $\kappa = 1.0$, the natural reward ($1436 \pm 96$) is much lower than other agents reported in Table 2.

## 5 CONCLUSION

In this paper, we first propose the optimal adversarial attack on state observations of RL agents, which is significantly stronger than many existing adversarial attacks. We then show the alternating training with learned adversaries (ATLA) framework to train an agent together with a *learned optimal* adversary to effectively improve agent robustness under attacks. We also show that a history dependent policy parameterized by a LSTM can be helpful for robustness. Our approach is orthogonal to existing regularization based techniques, and can be combined with state-adversarial regularization to achieve state-of-the-art robustness under strong adversarial attacks.

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

# A    APPENDIX

## A.1    FULL RESULTS OF ALL ENVIRONMENTS UNDER DIFFERENT TYPES OF ATTACKS

In Table 2, we only include the best attack rewards (lowest rewards over all attacks). In Table 3 we list the rewards under each specific attack. Note that, Robust Sarsa (RS) attack and our "optimal" policy attack both have hyperparameters. For RS attack we use the same set of 30 different settings of hyperparameters as in (Zhang et al., 2020b) to train a robust value function to attack the network. The reported RS attack result for each agent is the strongest one over the 30 trained value functions. For Snooping based attack, we use the "imitator" attack proxy as it was the strongest reported in (Inkawhich et al., 2019), and we attack every step of the agent. The imitator is a MLP or LSTM network according to agent policy network. We use the same loss KL divergence function as in the MAD attacks for this Snooping attack. We first collect state-action pairs for 100 episodes to train the "imitators", whose network structures are the same as the corresponding agents. In the test time, we first run MAD attack on it the "imitator" and then input the generated perturbed observation to the agent in a transfer attack fashion. For our "optimal" policy attack, the hyperparameters are PPO training parameters for the adversary (including the learning rate of the adversary policy network, learning rate of the adversary value network, the entropy regularization parameter and the ratio clip $\epsilon$ for PPO). We use a grid search of these hyperparameters to train an adversary that is as strong as possible, resulting in 100 to 200 adversaries produced for each agent. The reported optimal attack rewards is the lowest reward among all trained adversaries. Under this comprehensive adversarial evaluation, each agent is tested using hundreds of adversaries and the strongest adversary determines the true robustness of an agent.

Table 3: Average episode rewards $\pm$ standard deviation over 50 episodes on five baselines and SA-PPO (Zhang et al., 2020b). We report natural episode rewards (no attacks) and episode rewards under six adversarial attacks, including a simple random noise attack, the critic based attack in (Pattanaik et al., 2018), MAD and RS attacks in Zhang et al. (2020b), Snooping attack proposed in Inkawhich et al. (2019), and the optimal attack proposed in this paper. In each row we bold the best (lowest) attack reward over all five attacks. The row for the most robust method is highlighted.

| Env. | $\ell_\infty$ norm perturbation budget $\epsilon$ | Method | Natural Reward | Attack Reward Critic | Random | MAD | Snooping | RS | "Optimal" | Best Attack |
|---|---|---|---|---|---|---|---|---|---|---|
| Hopper | 0.075 | PPO (vanilla) | 3167±542 | 1464 ±523 | 2101±793 | 1410± 655 | 2234±1103 | 794±238 | **636± 9** | 636 |
| | | SA-PPO (Zhang et al., 2020b) | 3705± 2 | 3789± 15 | 2710± 801 | 2652± 835 | 2509±838 | 1130 ±42 | **1076± 791** | 1076 |
| | | Pattanaik et al. (2018) | 2755±582 | 2681± 555 | 2265± 502 | 1395±337 | 1349±436 | 1219± 174 | **291± 7** | 291 |
| | | ATLA-PPO (MLP) | 2559 ± 958 | 3497± 556 | 2153± 882 | 1679±676 | 1769±562 | 2329± 870 | **976± 40** | 976 |
| | | PPO (LSTM) | 3060± 639.3 | 2705± 986 | 2410± 786 | 2397± 905 | 2234±1103 | 811± 74 | **784± 48** | 784 |
| | | ATLA-PPO (LSTM) | 3487± 452 | 3524± 550 | 3474± 401 | 3081± 754 | 3130±692 | 1567± 347 | **1224± 191** | 1224 |
| | | ATLA-PPO (LSTM)+ SA Reg | 3291± 600 | 2073± 824 | 3165 ± 576 | 2814±725 | 2857±724 | 2244± 618 | **1772± 802** | 1772 |
| Walker2d | 0.05 | PPO (vanilla) | 4472 ± 635 | 3424 ± 1295 | 3007 ± 1200 | 2869 ± 1271 | 2786±962 | 1336 ± 654 | **1086±516** | 1086 |
| | | SA-PPO (Zhang et al., 2020b) | 4487± 61 | 4875± 30 | 4867± 39 | 3668± 1789 | 3928±1661 | 3808± 138 | **2908± 1136** | 2908 |
| | | Pattanaik et al. (2018) | 4058±1410 | 4058± 1410 | 2840± 2018 | 2927± 1954 | 2568±2044 | 1713 ±1807 | **733± 1012** | 733 |
| | | ATLA-PPO (MLP) | 3138 ± 1061 | 3243± 1004 | 3384 ± 1056 | 2596± 1005 | 2571±1084 | 3367± 1020 | **2213± 915** | 2213 |
| | | PPO (LSTM) | 2785± 1121 | 2730± 1082 | 2578 ± 1007 | 2471± 1109 | 2286±1156 | **1259± 937** | 1523± 869 | 1259 |
| | | ATLA-PPO (LSTM) | 3920± 129 | 3915± 274 | 3779 ± 541 | 3963 ± 36 | 3716±666 | **3219 ± 1132** | 3463± 1016 | 3219 |
| | | ATLA-PPO (LSTM) +SA Reg | 3842± 475 | 3884± 132 | 3927± 368 | 3836± 492 | 3742±629 | **3239± 894** | 3663± 707 | 3239 |
| Ant | 0.15 | PPO (vanilla) | 5687 ± 758 | 4934± 1022 | 5261± 1005 | 1759± 828 | 3668±547 | 268 ±227 | **-872 ± 436** | -872 |
| | | SA-PPO (Zhang et al., 2020b) | 4292± 384 | 4805 ± 128 | 4986 ±452 | 4662± 522 | 4079±768 | 3412 ±1755 | **2511 ± 1117** | 2511 |
| | | Pattanaik et al. (2018) | 3469± 1139 | 3469± 1139 | 2346± 459 | 1427± 625 | 1336±644 | 1289± 777 | **-672± 100** | -672 |
| | | ATLA-PPO (MLP) | 4894± 123 | 4427± 104 | 4541 ± 691 | 1891± 885 | 2862±1137 | 842± 143 | **33±327** | 33 |
| | | PPO (LSTM) | 5696 ± 165 | 5519 ± 114 | 5475 ± 691 | 3800± 363 | 3723±1168 | 1069 ± 382 | **-513 ± 104** | -513 |
| | | ATLA-PPO (LSTM) | 5612± 130 | 5196± 134 | 5390± 704 | 3903 ± 217 | 4455±677 | 1096± 329 | **716± 256** | 716 |
| | | ATLA-PPO (LSTM) +SA Reg | 5359 ±153 | 5295± 165 | 5366± 104 | 5240± 170 | 5135±413 | 4136± 149 | **3765± 101** | 3765 |
| HalfCheetah | 0.15 | PPO (vanilla) | 7117± 98 | 5761±119 | 5486 ± 1378 | 1836± 866 | 1637±843 | 489± 758 | **-660± 218** | -660 |
| | | SA-PPO (Zhang et al., 2020b) | 3632± 20 | 3589± 21 | 3619± 18 | 3624± 23 | 3616±21 | 3283± 20 | **3028 ±23** | 3028 |
| | | Pattanaik et al. (2018) | 5241± 1162 | 5440± 676 | 2910± 1694 | 1773± 1248 | 1465±726 | 1602± 1157 | **447± 192** | 447 |
| | | ATLA-PPO (MLP) | 5417± 49 | 5134± 38 | 5388 ± 34 | 4623 ±1146 | 4167±1507 | **2170± 2097** | 2709± 80 | 2170 |
| | | PPO (LSTM) | 5609± 98 | 4294± 112 | 5395± 158 | 4768± 106 | 4088±748 | 2899 ± 2006 | **-886± 30** | -886 |
| | | ATLA-PPO (LSTM) | 5766± 109 | 4008± 1031 | 5685± 107 | 4807± 154 | 4906±182 | 3458± 1338 | **2485± 1488** | 2485 |
| | | ATLA-PPO (LSTM) +SA Reg | 6157± 852 | 5991± 209 | 6164±603 | 5790± 174 | 5785±671 | **4806± 603** | 5058± 718 | 4806 |

## A.2    AGENT PERFORMANCE DURING TRAINING

In Table 3 we only report the agent performance at the end of training. In this subsection, we evaluate our agent performance during 20%, 40%, 60% and 80% of total training epochs using Robust Sarsa (RS) attacks. The results are presented in Table 4. The overall trend is that agents are getting stronger over time ("RS attack reward" is increasing), achieving better robustness in later checkpoints.

## A.3    NETWORK STRUCTURE

For fully connected networks, we use the same network as in (Zhang et al., 2020b), which contains 2 hidden layers with 64 hidden neurons each layer, for both policy and value networks, for both the

Table 4: Natural and RS attack rewards of ATLA-PPO (LSTM)+ SA Reg checkpoints during training. We report Average rewards± standard deviation over 50 episodes.

| Environment | Reward | 20% | 40% | 60% | 80% | 100% |
|---|---|---|---|---|---|---|
| Hopper | Natural Reward | 3440±11 | 1161 ±485 | 3013±584 | 3569±161 | 3291±600 |
| | RS Attack Reward | 716±82 | 631±51 | 1089±501 | 3181±634 | 2244±618 |
| Walker2d | Natural Reward | 989±254 | 3506±174 | 2203±988 | 3803±726 | 3842±475 |
| | RS Attack Reward | 882±269 | 1744±347 | 739±531 | 2550±1020 | 3239±894 |
| Ant | Natural Reward | 2634±1222 | 4532±106 | 5007±143 | 5127±542 | 5393±139 |
| | RS Attack Reward | 216±171 | 1903±93 | 3040±241 | 3040±241 | 4136±149 |
| HalfCheetah | Natural Reward | 4525±140 | 5567±138 | 5955±177 | 5956±181 | 6300±261 |
| | RS Attack Reward | 3986±564 | 3986±564 | 4911±923 | 4571±1314 | 4806±603 |

agent and the adversary. For LSTM agents, we use a single layer LSTM with 64 hidden neurons, along with an input embedding layer projecting state dimension to 64 and an output layer projecting 64 to output dimension. For LSTM agents, when conducting the "optimal" attack, we also use a LSTM network for the adversary to ensure the adversary is powerful enough.

## A.4 HYPERPARAMETER FOR THE LEARNING-BASED "OPTIMAL" ATTACK

Our "optimal" attacks uses policy gradient methods to learn the optimal adversary during agent testing, and each learning process involves the selection of hyperparameters. Specifically, the hyperparameters include the learning rates of the adversary's policy and value networks, the entropy coefficient, and the annealing of the learning rate. To reduce search space, for ATLA agents, the learning rates of the testing phase adversary's policy and value networks are chosen ranging from 0.3X to 3X of the learning rates of adversary's policy and value networks used in training. For other agents trained without an adversary, the learning rates of the testing phase adversary's policy and value networks are chosen ranging from 0.3X to 3X of the learning rates of the agent's policy and value networks. We tested both linearly annealed learning rate and non-annealing learning rate. The adversary's entropy coefficient is chosen form 0 and 0.003. The final results reported in all tables are the best (lowest) reward achieved by the "optimal" attacks among all hyperparameter configurations. Typically this includes around 100 to 200 different adversaries trained with different hyperparameters. This guarantees the strength of this attack and allows a comprehensive evaluation of the robustness of all agents.

## A.5 HYPERPARAMETERS FOR ATLA PERFORMANCE EVALUATION

**Hyperparameters for PPO (vanilla)** For the Walker2d and Hopper environment, we use the same set of hyperparameters as in (Zhang et al., 2020b); the hyperparameters were originally from (Engstrom et al., 2020) and found using a grid search experiment. We found that this set of hyperparameters work well. For HalfCheetah and Ant environment, we use a grid search of hyperparameters, including the learning rate of the policy network, learning rate of the value network and the entropy bonus coefficient. For Hopper, Walker2d and HalfCheetah environments, we train for 2 million steps (2 million environment interactions). For Ant, we train for 10 million steps. Training for longer may slightly improve agent performance under no attacks, but has no impact for performance under strong adversarial attacks.

**Hyperparameters for PPO (LSTM)** For PPO (LSTM), we conduct a smaller scale hyperparameter search. We search hyperparameter values that are close to the optimal ones found for the PPO vanilla agent. We train these LSTM agents for the same steps as those in vanilla PPO.

**Hyperparameters for SA-PPO** We use the same value for all hyperparameters as in vanilla PPO except SA-PPO's extra $\kappa$ for the strength of SA-PPO regularization. For $\kappa$, we choose from $1 \times 10^{-6}$ to 1. We train agents with each $\kappa$ 21 times and choose the $\kappa$ value whose median agent has the highest worst-case reward under all attacks.

**Hyperparameters for ATLA-PPO** For ATLA-PPO, we have hyperparameters for both agent and adversary. We keep all agent hyperparameters the same as those in vanilla MLP/LSTM agents, except for the entropy bonus coefficient. We find that sometimes we need a larger entropy bonus co-

Table 5: Hyperparameters for all environments and settings. For vanilla environments, we use the hyperparameters from Zhang et al. (2020b) and Engstrom et al. (2020) if they are available for that environment (Hopper and Walker2d). Other environments' hyperparameter for the vanilla PPO model is found by a grid search. For SA-PPO and ATLA-PPO (MLP), the same set of hyperparameters as in the vanilla models are used, except that for SA-PPO we tune the parameter $\kappa$ and for ATLA-PPO (MLP) we tune the entropy bonus coefficients as well as learning rates for the adversary. For LSTM models, we first tune the vanilla LSTM PPO models and find the best learning rates, keep using them in all LSTM based models.

| Env. | model | policy lr | val lr | entropy coeff. | $\kappa$ | adv. policy lr | adv. val lr | adv. entropy coeff. |
|---|---|---|---|---|---|---|---|---|
| Hopper | PPO(vanilla) | 3e-4 | 2.5e-4 | 0 | – | – | – | – |
| | SA-PPO | 3e-4 | 2.5e-4 | 0 | 0.03 | – | – | – |
| | ATLA-PPO (MLP) | 3e-4 | 2.5e-4 | 0.01 | – | 0.001 | 0.0001 | 0.001 |
| | PPO (LSTM) | 1e-3 | 3e-4 | 0.0 | – | – | – | – |
| | ATLA-PPO (LSTM) | 1e-3 | 3e-4 | 0.01 | – | 0.01 | 0.01 | 0.001 |
| | ATLA-PPO (LSTM)+ SA Reg | 1e-3 | 3e-4 | 0.01 | 0.3 | 0.003 | 0.01 | 0.003 |
| Walker2d | PPO(vanilla) | 4e-4 | 3e-4 | 0 | – | – | – | – |
| | SA-PPO | 4e-4 | 3e-4 | 0 | – | – | – | – |
| | ATLA-PPO (MLP) | 4e-4 | 3e-4 | 0.0003 | – | 0.0001 | 0.0001 | 0.002 |
| | PPO (LSTM) | 1e-3 | 3e-2 | 0 | – | – | – | – |
| | ATLA-PPO (LSTM) | 1e-3 | 3e-2 | 0.001 | – | 0.0003 | 0.03 | 0 |
| | ATLA-PPO (LSTM)+ SA Reg | 1e-3 | 3e-2 | 0.001 | 0.3 | 0.003 | 0.03 | 0.001 |
| Ant | PPO(vanilla) | 5e-5 | 1e-5 | 0 | – | – | – | – |
| | SA-PPO | 5e-5 | 1e-5 | 0 | 3e-3 | – | – | – |
| | ATLA-PPO (MLP) | 5e-5 | 1e-5 | 3e-4 | – | 1e-05 | 3e-06 | 0 |
| | PPO (LSTM) | 3e-4 | 3e-4 | 0 | – | – | – | – |
| | ATLA-PPO (LSTM) | 3e-4 | 3e-4 | 0.0003 | – | 0.0003 | 0.0001 | 0.0003 |
| | ATLA-PPO (LSTM)+ SA Reg | 3e-4 | 3e-4 | 0.003 | 0.1 | 0.0003 | 3e-05 | 3e-05 |
| HalfCheetah | PPO(vanilla) | 3e-4 | 1e-4 | 0 | – | – | – | – |
| | SA-PPO | 3e-4 | 1e-4 | 0.1 | – | – | – | – |
| | ATLA-PPO (MLP) | 3e-4 | 1e-4 | 0.0003 | – | 0.001 | 0.0003 | 0.003 |
| | PPO (LSTM) | 1e-3 | 3e-4 | 0 | – | – | – | – |
| | ATLA-PPO (LSTM) | 1e-3 | 3e-4 | 0.0003 | – | 0.003 | 0.001 | 0 |
| | ATLA-PPO (LSTM)+ SA Reg | 1e-3 | 3e-4 | 0 | 0.03 | 0.003 | 0.003 | 0.0003 |

efficient in ATLA to allow sufficient exploration of the agent, as learning with an adversary is harder than learning in attack-free environments. For the adversary, we run a small-scale hyperparameter search on the learning rate of adversary policy and value networks, and the entropy bonus coefficient for the adversary. To reduce the number of hyperparameters for searching, we use values close to those of the agent. We set $N_\nu = N_\pi = 1$ in all experiments and did not tune this hyperparameter. For ATLA, we train 5 million steps for Hopper, Walker and HalfCheetah and 10 million steps for Ant. We find that similar to the observations in (Madry et al., 2018), training with an adversary typically requires more steps to converge, however in all our environments the training does reliably converge.

**Agent selection** For each setup, we repeat the experiments using the same set of hyperparameters for 21 times due to the high performance variance in RL. We then attack all the agents using random, critic, MAD and RS attacks. We use the lowest reward among all attacks as a metric to rank those agents. Then, we select the agent with *median* robustness as our final agent. This final agemt is then attacked using the "optimal" attack to further reduce its reward. The numbers we report in Table 2 are not from the best runs, but the runs with median robustness. This is done to improve reproducibility as RL training process can have high variance.

