# OpenReview forum: "Robust Reinforcement Learning on State Observations with Learned Optimal Adversary"
_ICLR.cc/2021/Conference — ICLR 2021 Poster_

### Official Review · AnonReviewer4 · 2020-10-28
**This paper has good insights, but more work is needed**

**Rating:** 6
**Confidence:** 5

**Review:**

Summary:
This paper proposes to improve the robustness of a reinforcement learning agent by alternatively training an agent and an adversary who perturbs the state observations. The learning of an “optimal” adversary for a fixed policy is based on the theory of SA-MDP in prior work. The learning of an optimal policy under a fixed adversary is done by solving a POMDP problem. Experimental results show that the proposed alternating training with learned adversaries (ATLA) framework can improve the performance and robustness of PPO.

Strengths:
1. The paper is well-organized and easy to follow.
2. The authors distinguish between the vulnerability of function approximations and the intrinsic weakness of policy, which is interesting and can be useful for the community to investigate the vulnerability of deep RL.
3. The experiment results show that both the learned adversary and the trained agent perform well, in terms of attacking and learning respectively. In addition, the proposed LSTM-based policy is shown to be more robust than regular feedforward NNs.

Weakness:
1. The novelty of this paper is a little limited. (1) The idea of alternative training the agent and the adversary is similar to RARL[1], and Algorithm 2 (ATLA) is similar to Algorithm 1 in [1]. Although RARL focuses on the case where adversary directly changes the environment and ATLA focuses on the observation perturbation attacks, the whole ideas are still similar. (2) The main method of learning an adversary is based on the theoretical work of SA-MDP[2].
2. The authors claim that the proposed adversary is strong since it follows the theoretical framework of SA-MDP. However, in Lemma 1, the adversary reward in SA-MDP, \hat{R}, is defined as the weighted average of -R, while in Algorithm 1, the adversary reward is given by -R itself. It is not clear why such a relaxation still follows the theoretical framework. More details illustrating the approximation and some analysis about the optimality will be appreciated.
3. In the experiment section, the authors only compare the proposed algorithm with [2] in terms of “optimal” attack and robust training. However, there are a lot of works that attack the observations of a fixed policy [3,4,5]. And more importantly, [6] also proposes to train a robust agent under adversarial attacks. It will be more convincing if the authors empirically or theoretically compare with some of these potential baselines.
4.The computational complexity / sample complexity of the proposed ATLA might be problematic, as for each iteration of learning, the adversary needs to solve a new MDP, which makes the proposed robust training less practical to use.

Minor comments:
- This paper sometimes uses the phrase "training time attack" to refer adversarial attacks, which is misleading, e.g. the second contribution, the third paragraph of related work. Training-time attack usually refers to poisoning attack, which changes the training dataset and alters the learned policy, different from the scenario in this paper where an adversary wants to fool a fixed policy.


Refs:
[1] Pinto, Lerrel, et al. "Robust adversarial reinforcement learning." arXiv preprint arXiv:1703.02702 (2017).
[2] Zhang, Huan, et al. "Robust Deep Reinforcement Learning against Adversarial Perturbations on Observations." arXiv preprint arXiv:2003.08938 (2020).
[3] Huang, Sandy, et al. "Adversarial attacks on neural network policies." arXiv preprint arXiv:1702.02284 (2017).
[4] Lin, Yen-Chen, et al. "Tactics of adversarial attack on deep reinforcement learning agents." arXiv preprint arXiv:1703.06748 (2017).
[5] Inkawhich, Matthew, Yiran Chen, and Hai Li. "Snooping Attacks on Deep Reinforcement Learning." arXiv preprint arXiv:1905.11832 (2019).
[6] Pattanaik, Anay, et al. "Robust deep reinforcement learning with adversarial attacks." arXiv preprint arXiv:1712.03632 (2017).

---

> ### Author Response · Authors · 2020-11-17
> **[2/2] We added new attacks and baseline results, and further explained Lemma 1**
>
> 3. Additional experiments and comparisons:
>
>     3.1 Additional Attacks:
> Among the mentioned attacks, [3] is one step FGSM attack using the value function, which is weaker than the multi-step PGD attack proposed in [6]. Our paper already includes the stronger multi-step attack in [6] (named as “critic attack” )
> [4] proposed “strategically-timed attack” and the “enchanting attack”. The strategically-timed attack used a multi-step gradient based attack similar to [6] for attacking only partial frames to avoid detection, so this is a weaker attack with more constraints for the attacker. In our paper, all attacks are applied in every step, which is stronger than the threat model used in [4]. The goal of the enchanting attack is to lure the agent into a certain state, and it is not directly minimizing cumulative reward. Since our goal is to reveal the agent’s true robustness, we always evaluate the agent under the strongest possible attack that reduces agent reward most, so we did not use attacks in [4].
> [5] proposed “snooping attack”, which is a black-box adversarial attack. As suggested by the reviewer, we implemented Snooping attack in [5] and tested it on all our agents. The results are shown in Table 1 (main text) and Table 3 (appendix). From the results, we can see that Snooping attack has strength similar to the MAD attack, and is typically worse than RS attack and the “optimal” attack proposed in this paper. Thus, the main conclusion and evaluation results do not change after adding this new attack into comparison.
>
>     3.2 Comparison to [6] for robust training under attack:
> As suggested by the reviewer, we added a comparison to [6], which uses the critic attack for adversarial training. The results are included in Table 2 (main text) and Table 3 (appendix). We find that this method cannot reliably improve robustness under our suite of strong attacks. The reason is that the critic attack is a relatively weak adversary, so the agent learned with a weak adversary cannot defend against a stronger adversary.
>
> 4. Sample complexity:
> We want to emphasize that just as in supervised learning settings (e.g., image classification), adversarial training can significantly increase computational complexity, e.g., [7] must be trained with more epochs than ordinary training to converge. There is no free lunch for adversarial robustness [8]. Our proposed method usually requires up to 5X more iterations compared to vanilla PPO agents, which is comparable to adversarial training for supervised learning. We believe the computational complexity of our method is reasonable, especially considering that the field of adversarial robustness in DRL is relatively new. Further reducing the computational or sample complexity of adversarial training in RL setting is a good future direction.
>
> 5. Phrase "training time attack"：
> Thank you for pointing out this potential confusion. We have changed "training time attack" to "adversarial training" in our paper and made it clear that we use adversarial attacks on state observations to improve the robustness of the agent during test time.
>
> Conclusion:
> We hope Reviewer 4 can re-evaluate our paper based on our new empirical results (snooping attacks [5] and an adversarial training baseline [6]) and clarifications on our Lemma, Algorithm, and sample complexity. Thank you and we will be glad to answer any additional questions you may have.
>
> References:
> [1-6]: the same as the references in your review.
> [7] Madry, Aleksander, et al. "Towards deep learning models resistant to adversarial attacks." arXiv preprint arXiv:1706.06083 (2017).
> [8] Schmidt, Ludwig, et al. "Adversarially robust generalization requires more data." Advances in Neural Information Processing Systems. 2018.

---

> ### Author Response · Authors · 2020-11-17
> **[1/2] We added new attacks and baseline results, and further explained Lemma 1**
>
> We really appreciate your helpful comments. We have added additional experiments to include the new attack you mentioned, and also compared with existing adversarial training methods. We address your concerns below.
> 1. Novelty:
> First, although the idea of learning an optimal adversary is from a lemma in SA-MDP[2], [2] did not use it to train an adversary for improving agent robustness, and also did not reveal the POMDP nature of learning a robust agent (our Lemma 2); based on this insight, we show the necessity of using non-Markov policies for robust agents with LSTM. No prior works in this area pointed out this connection to POMDP or proposed non-Markov policies for adversarial robustness. Our ATLA framework outperforms existing works using pure regularization [2] and adversarial training with weak adversaries [6], sometimes by a large margin, and achieves state-of-the-art performance.
> Second, as you pointed out, our problem is fundamentally different from RARL[1], as RARL focuses on environment changes and we focus on perturbations on state observations. Indeed, learning an adversary has been used as a key idea in many works such as GANs, but our work is the first to follow a solid theoretical framework (SA-MDP) to analyze the learning procedure of the agent and the adversary in the setting of perturbations of observations in RL setting. Importantly, unlike RARL, the agent learning problem is a POMDP in our case.
> Lastly, we empirically demonstrate that the “optimal” adversarial attack on DRL agents is very effective, as this attack is backed by solid theory while most existing attacks[3,4,5] are based on certain heuristics. In Figure 3 and Table 1 we show that this attack is significantly stronger than previous attacks. It can become a strong benchmark for evaluating the robustness of DRL agents in future works.
>
> 2. Lemma 1 and Algorithm 1:
> Thank you for pointing this out. Algorithm 1 and Lemma 1 actually do match, and we have updated our paper to further explain it.
> Specifically, in Lemma 1, the reward function $\hat{R}(s, \hat{a}, s’)$ by definition is a conditional expectation of the adversary’s reward. The adversary’s reward is a random variable. See our updated Lemma 1 which includes this expectation explicitly.  In Algorithm 1, during agent rollout, $\hat{r} = -r$ is just one sample of this random variable. Formally, we include the derivation of the distribution of $p(\hat{r} | s, \hat{a}, s’)$ and its expectation in Appendix A, and you will find that when we assign the adversary reward $-r$ when the agent’s reward is $r$, you will get the expectation in Lemma 1. So there is no discrepancy between Lemma 1 and Algorithm 1.

---

> ### Comment · AnonReviewer4 · 2020-11-17
> **Update of my review**
>
> Thank you for the detailed response. The additional explanations and experiments have addressed most of my concerns, so I will increase my score to 6.

---

### Official Review · AnonReviewer1 · 2020-10-28
**A well written and motivated paper that training agents and adversary alternatively**

**Rating:** 7
**Confidence:** 5

**Review:**

The paper is very well written and the considered problem of training an adversary along with the agent is very interesting. Within the proposed concept, the parameterized adversary can be trained by viewing the agent as a part of the environment, so it avoids to access the parameters of the agent policy. From the perspective of the agent, with an unknown adversary, the MDP becomes a POMDP with uncertainty hidden in the adversary, and hence the fact of using LSTM policy is much better for the agent is reasonable. The entire problem is wisely formulated.  The experimental settings are well designed and results support the positiveness of the proposed framework.

I only have one comment that the SA-MDP can also be understood from another perspective. That is, SA-MDP is actually an asymmetric competitive multi-agent problem, and the alternative training of agent and adversary can be viewed as an instance of self-play. Also, the optimality of SA-MDP for either the agent or the adversary can be explained through multi-agent RL or game theory. It would be interesting if the authors could take a look into such a direction.

---

> ### Author Response · Authors · 2020-11-17
> **Thank you for the encouraging comments!**
>
> Thank you for the great summarization of our main contributions and we really appreciate your encouraging comments. The perspective of understanding SA-MDP from asymmetric competitive multi-agent problems is insightful. We will study further into this direction to connect asymmetric game theory to SA-MDP. Thank you for providing this great insight! Feel free to let us know if you have any further questions regarding our paper.

---

### Official Review · AnonReviewer2 · 2020-10-29
**Official Blind Review #2**

**Rating:** 7
**Confidence:** 3

**Review:**

Summary of the paper:  The paper studies adversarial attacks in RL, focusing both on the design of optimal attack strategies on RL agents, as well as robust training RL procedures for mitigating attacks. Building on the results of (Zhang et al., 2020), the paper proposes a new learning framework (ATLA), that simultaneously trains a (strong) adversary and a (robust) deep RL agent. The paper showcases the importance of the new framework through extensive experimental evaluation.

Reasons for score: Overall, I find the paper to be an interesting read and its contribution relevant to the line of work on adversarial attacks in RL. The contributions of the paper seem non-trivial, and include a framework for designing optimal attacks and training procedures that can optimize for robustness. As shown by the experiments, the proposed solution leads to significant increase in performance compared to state of the art baselines. These results complement those of (Zhang et al., 2020). Nonetheless, the presentation of the paper could be improved in terms of clarity. Some parts of the paper could be reorganized and explained in more detail. Suggestion for improvements and questions are outlined below.

Clarity: The paper is overall enjoyable to read, but some parts are not clearly/precisely written. There are quite a few typos, some of which might be important for understanding the content. I did not follow equation (2), which seems to contain typos. Could you explain in more detail the loss function defined with this equation? Furthermore, notation in the paragraph before section 3.1 is partly confusing, in particular, $\nu$ seems to be a deterministic function, but then for observation $\hat s$ we have $\hat s \sim \nu(s)$ indicating that $\nu(s)$ might be a distribution from which we sample $\hat s \sim \nu(s)$. I would also suggest reorganizing the content related to Figure 1, which is introduced in section 1, but only explained in detail in section 3.1.

Related Work: The related work is generally covered well, but it could be expanded by providing some connection to the line of work that studies policy teaching and policy poisoning attacks in RL. Could you explain how the setting of this paper compares to those studied in this line of work (e.g., Parkes et al. 'Policy Teaching Through Reward Function Learning', Ma et al. 'Policy Poisoning in Batch Reinforcement Learning and Control', Rakhsha et al., 'Policy Teaching via Environment Poisoning: Training-time Adversarial Attacks against Reinforcement Learning', etc.)?

Experiments: I'm wondering to what extent are the results for different methods in Table 2 comparable. Namely, the methods seem to be based on different architectures, so it is not immediately clear what conclusions should be drawn from these results. Surprisingly, the discussion on page 8 does not seem to compare the results of ATLA-PPO (MLP) and SA-PPO. Could you elaborate more on these results and make relative comparison? Furthermore, in the following sentence: 'For reproducibility, for each setup we train at least 5 (up to 21) agents, attack all of them and report the one with median robustness.' - why is there discrepancy between different setups in the number of trained agents?

---

> ### Author Response · Authors · 2020-11-17
> **Thank you for the comments and suggestions! Please see our response below.**
>
> We greatly appreciate the helpful comments from the reviewer. They help us improve our paper a lot. We now answer the reviewer’s questions below:
>
> 1. Equation (2):
> Thank you for pointing this out. We have updated Eq. 2 (now becomes Eq. 3), fixed the typos, and added more explanations.
>
> 2. Deterministic or stochastic $\nu$:
> Sorry for mixing notations here. We have updated our paper and now we keep using the stochastic adversarial notation.
>
> 3. Figure 1:
> We have improved the caption below Figure 1 to make this example more self-contained and also added pointers to later sections in Figure 1. We hope this figure can be a good example in the Introduction to explain that the robustness issue is not just in deep reinforcement learning with function approximation.
>
> 4. Policy teaching and policy poisoning:
> We really appreciate the provided references and we have cited them in the related work section. Policy teaching and policy poisoning manipulate the reward or cost signal during agent training time to induce the desired agent policy.  Essentially, policy teaching is a training time “attack” with perturbed rewards from the environments (which can be analogous to data poisoning attacks in supervised learning settings), while our goal is to train a robust agent against test time (not training time) adversarial attacks on state observations (not rewards).
>
> 5. Discussions in experiments:
> Sorry for the confusion. We included both MLP and LSTM agents because in Section 3.2, we show that learning an agent under adversary is a POMDP problem, so a history-dependent policy (LSTM policy) can potentially perform better. We observe that ATLA (LSTM) overall outperforms ATLP (MLP) in experiments, matching our theoretical observation in Section 3.2. We have updated the paragraph on discussion experiment results to cover all methods, and structured this paragraph to clearly distinguish between MLP and LSTM models.
>
> 6. Discrepancy in the number of trained agents
> This is mainly due to computational constraints. To ensure reproducibility, we train each setting multiple times, and each agent is evaluated by hundreds of independent adversaries (because some attacks have hyperparameters, and we run a grid search to train a large number of adversaries). We report the agent with median robustness overall repeated training runs. The computation cost for training and attacks is high so we were not able to repeat all settings by 21 times.
> During the discussion period, we trained more agents and now almost every setting has 21 agents. We find that the median reward under attack is roughly the same as the ones reported in our paper, so our main results are reproducible and remain unchanged.
>
> We thank the reviewer again for the very helpful comments, and please feel free to let us know if any of your concerns are still not addressed or if you have further questions.

---

### Official Review · AnonReviewer3 · 2020-10-30
**An interesting framework for robust DRL**

**Rating:** 7
**Confidence:** 2

**Review:**

The authors presents how to learn optimal adversary following the state-adversarial Markov decision process (SA-MDP), and also proposes alternating training with learned attacks (ATLA) framework that trains the optimal adversary online together with the agent to improve the robustness of the DRL agent. Experiment result shows that ATLA outperforms the explicit regularization based methods.

Overall I think the paper is well-written and clearly illustrated the methodology. The experimental results are mostly comprehensive. The contribution is clear. I still have a few questions and concerns below:

1) It is a natural and interesting idea to use alternating training to optimize both the adversary and the agent online. In terms of finding the (near) optimal adversary following the theory of SA-MDP, the authors argue that because of using "function approximator to learn the agent so it’s no longer optimal". However, it seems unclear how much the approximation affects the optimality. For tabular case instead of DRL, is it possible to really find the optimal adversary, and if so, how? How far away the learned strong adversary from the real optimal one? Indeed from experiments the learned adversary can better attack than other baselines, but it would still be important to understand the advantage and room to improve in a principled way. For finding the policy, it is understandable that because of the difficulty of solving POMDP, this paper does not solve the optimal policy.

2) ATLA-PPO + SA Reg further improves performance over ATLA-PPO in experiments. This seems to suggest that the advantage of ATLA-PPO and SA Reg are complementary. Does the regularization on the function approximator provides additional robustness or it covers the error of function approximator using LSTM? Is it possible to understand this in experiments?

3) Are the adversary and the agent both getting stronger over time? The paper only showed final results and did not show the running time result. Hypothetically because of the alternating training, the adversary and the agent should both be improved and it would be interesting to verify this in experiments.

---

> ### Author Response · Authors · 2020-11-17
> **Thank you for your questions! Please see our response below**
>
> We really appreciate the detailed review comments and we provide our response below.
>
> 1. Optimality under approximation
>
> This is a very good question, but unfortunately, it is a very challenging question and beyond the reach of our paper. In section 3.1, we formulate the learning problem of an optimal adversary as finding an optimal policy on MDP. Thus, finding the optimal adversary itself becomes a general RL problem. When a deep neural network function approximator is used to solve an RL problem, currently very limited theoretical study is done to show the gap between the learned policy and the optimal one, and in some cases, even the optimal policy itself is beyond our reach so it is very hard to compare the two. For example, although we can combine RL with function approximators to solve the game of Go and outperform top-level human players, it is unclear to us what is the optimal policy for Go and how far our currently learned agents are away from it.
>
>
> 2. Improvement from SA Reg
>
> As the reviewer correctly pointed out, ATLA and SA-reg are two complementary methods. Firstly, the SA regularization helps model robustness, as Theorem 5 suggested in Zhang et al. Secondly, as a regularization, SA-reg makes function approximators more smooth so they become more robust under small perturbations and have less “errors”. Both of the two factors are effective and it is hard to distinguish them from each other. When combined with our ATLA framework, we believe the smoothness caused by the regularizer might be more important, as our ATLA framework does not explicitly encourage smoothness. In high dimensional environments like Ant where smoothness is more critical, we can see the ATLA with SA-reg works more effectively than the ones without SA-reg. We added this discussion in Section 4.
>
> 3. Adversary and agent get stronger over time
>
> We provide additional results on evaluating agents during training. For our best setting (ATLA LSTM + SA reg) we evaluate the model checkpoints with RS attack at 20%, 40%, 60%, 80% training steps. The results are provided in Table 4 in Appendix A.3. The overall trend is that agents are getting stronger over time, achieving better robustness in later checkpoints.
>
> It is unsure how to fairly evaluate the adversary during training since intermediate adversaries are closely related to their corresponding training agents. Similarly, it does not quite make sense to evaluate the accuracy of the discriminator in GAN training, as the discriminator is specific to the generator under training, and what we care about is the performance of the generator (analogous to the agent in our setting). We can access the quality of the adversary by evaluating the robustness of the agent - if the agent is robust to strong adversarial attacks such as RS attacks, the adversary should also be strong, otherwise, the agent cannot learn to be robust. We included an evaluation of the agents in Appendix A.3, as discussed above.
>
>
> We thank you again for your helpful comments and please let us know if you have any additional questions or concerns.

---

### Decision · Program_Chairs · 2021-01-07
**Final Decision**

**Decision:**

Accept (Poster)

**Comment:**

The paper describes a new technique to train an adversarial MDP to perturb the observations provided by the environment.  This adversarial MDP is then used to train an RL agent to be more robust.  Since the adversarial agent essentially defines an observation distribution for the environment, the RL agent needs to optimize a POMDP.  This is nice work that was unanimously praised by the reviewers.  It produces stronger adversaries and more robust RL agents than previous work.  This represents an important contribution to the state of the art of robust RL.